# "He who pays the piper calls the tune": Researcher experiences of funder suppression of health behaviour intervention trial findings

Sam McCrabb[1]*, Kaitlin Mooney[1], Luke Wolfenden[1,2], Sharleen Gonzalez[1], Elizabeth Ditton[1], Serene Yoong[1,2,3], Kypros Kypri[1]

1 School of Medicine and Public Health, Faculty of Health and Medicine, University of Newcastle, Callaghan, New South Wales, Australia, 2 Hunter New England Population Health, Hunter New England Local Health District, Wallsend, New South Wales, Australia, 3 School of Health Sciences, Swinburne University of Technology, Hawthorn, Vic, Australia

* sam.mccrabb@newcastle.edu.au

## Abstract

### Background

Governments commonly fund research with specific applications in mind. Such mechanisms may facilitate 'research translation' but funders may employ strategies that can also undermine the integrity of both science and government. We estimated the prevalence and investigated correlates of funder efforts to suppress health behaviour intervention trial findings.

### Methods

Our sampling frame was lead or corresponding authors of papers (published 2007–2017) included in a Cochrane review, reporting findings from trials of interventions to improve nutrition, physical activity, sexual health, smoking, and substance use. Suppression events were based on a previous survey of public health academics. Participants answered questions concerning seven suppression events in their efforts to report the trial, e.g., [I was. . .] "asked to suppress certain findings as they were viewed as being unfavourable." We also examined the association between information on study funder, geographical location, targeted health behaviour, country democracy rating and age of publication with reported suppression.

### Findings

We received responses from 104 authors (50%) of 208 eligible trials, from North America (34%), Europe (33%), Oceania (17%), and other countries (16%). Eighteen percent reported at least one of the seven suppression events relating to the trial in question. The most commonly reported suppression event was funder(s) expressing reluctance to publish because they considered the results 'unfavourable' (9% reported). We found no strong

**Funding:** Professor Wolfenden is funded by a NHMRC Career Development Fellowship (APP1128348) and a Heart Foundation Future Leader Fellowship (Award Number 101175). Infrastructure funding was provided by Hunter New England Population Health and The University of Newcastle. Professor Kypri was funded by a University of Newcastle Brawn Senior Research Fellowship for his input to this study. A/Prof Serene Yoong is funded by an ARC Discovery Early Career Researcher Award (DE170100382).

**Competing interests:** The authors have declared that no competing interests exist.

associations with the subject of research, funding source, democracy, region, or year of publication.

## Conclusions

One in five researchers in this global sample reported being pressured to delay, alter, or not publish the findings of health behaviour intervention trials. Regulation of funder and university practices, establishing study registries, and compulsory disclosure of funding conditions in scientific journals, are needed to protect the integrity of public-good research.

## Introduction

Generating scientific knowledge should be, in principle, a key consideration in the design of programmes to improve public health. Governments fund national agencies whose purpose is supporting science (e.g., the National Institutes of Health in the United States of America [USA], National Health and Medical Research Council (NHMRC) [Australia]), and researcher-initiated projects are routinely funded through such agencies. Research funding is also dedicated to addressing the priorities of funders with objectives typically relating to informing public policy or commercial imperatives [1]. Such strategic funding aims to address knowledge gaps important to funders thereby facilitating 'research translation' by ensuring relevance to end-users. However, these funding models have been shown to undermine the integrity of science by enabling funders to influence how research is done and reported [2–4].

As providers of publicly funded health and medical research, universities have a vital role in facilitating independent enquiry. The notion of academic freedom is that researchers bound by the scholarly conventions of peer review and ethical approval, are free to do research without interference or the threat of professional disadvantage [5]. Many see the preservation of such freedom as vital to safeguarding the reflection, critique, and innovation that academia can bring to society [3, 6]. However, academic integrity is increasingly undermined by the influence of vested interests on research [2, 7], and a reproducibility crisis [8], calling into question whether public research institutions actually serve the public interest [9]. That research funders who are also responsible for giving policy advice or implementing intervention programmes have a stake in study findings, puts pressure on the impartiality of the researchers who depend on the funding. This could include subtle pressure on researchers, unconsciously conveyed hopes for 'positive' findings, or total suppression or censorship of reports for political advantage [4]. Various mechanisms exist to regulate researcher behaviour, including codes of conduct and ethical review [10, 11]. In Australia, many government funding agreements require researchers to obtain funder approval to publish reports [12].

The suppression of public-good research by funders or other parties is neither well understood nor coherently regulated [13]. A 2006 survey of Australian public health researchers reported that 21% of participants (with a response rate of 46%) had experienced at least one incident in which a government funder suppressed their research in the preceding 5.5 years [14]. The most common forms of suppression reported were blocking, significantly delaying publication and requests to "sanitise" reports [14]. A survey of Australian ecologists and conservation scientists by Driscoll et al. [15] indicated that Government and industry respondents reported higher rates of suppression than university respondents (34%, 30% and 5% respectively) and this suppression was mainly in the form of internal communication and media. A 2015 Canadian survey of federal government scientists showed that within the last 5 years 24%

of scientists had been asked to exclude certain findings from their reports, and 37% reported that they were prevented from responding to media enquiries within their area of expertise [16]. In the United Kingdom (UK) an enquiry into public-good research, commissioned by a science charity, presented nine case studies outlining the impact of significant delays in the publication of findings. In several cases delays appeared to be motivated by political considerations [4]. Knowing how often and in what circumstances the suppression of public health research occurs is important because of the potential impact of withholding, delaying, or misrepresenting findings. This is acutely apparent in the COVID-19 pandemic, where delays in releasing early research findings in China allowed significant outbreaks to occur in other countries [17–19].

The aims of this study were: (1) to ascertain the reported prevalence of efforts to suppress the findings of primary prevention trials that target nutrition, physical activity, sexual health, tobacco use, alcohol or substance use; and (2) to identify associations between trial characteristics and suppression events.

## Methods

### Design

We invited the lead authors of primary prevention trials included in Cochrane reviews to complete a Computer Assisted Telephone Interview or online survey. This study was part of a larger cross-sectional study that investigated researchers' experiences in developing, conducting and evaluating public health interventions, the effectiveness of the intervention, any knowledge translation strategies used, and reported impacts on health policy and practice (unpublished). The present study investigates the prevalence of suppression of trial findings, and how these relate to the trial characteristics. The University of Newcastle Human Research Ethics Committee approved the study protocol (H-2014-0070). Completion of the online survey was taken as implied consent.

### Sampling

We searched the Cochrane Library for reviews that were: (1) focused on primary prevention or included trials with setting-based primary prevention components; and (2) related to nutrition, physical activity, sexual health, tobacco use, alcohol use, or other psychoactive substance use. These risk behaviour areas were chosen as the larger cross-sectional study this sub-study is a part of was interested in these health behaviours.

We classified primary studies from the reviews as eligible if they were randomised controlled trials (RCT) or non-randomised controlled trials investigating the effects of efforts to modify nutrition, physical activity, sexual health, tobacco use, alcohol use, or other substance use. We limited eligibility to English language reports published from 2007–2017.

### Recruitment and data collection

Authors from identified articles were invited to participate if they were one of the first two authors, the last author, or the corresponding author. Contact information was sourced from the public domain. We contacted corresponding authors first to complete the survey on behalf of all authors. Corresponding authors could nominate co-authors to complete the survey on their behalf. If after four week we had no response to the corresponding author, we invited the first, second and/or last author of the trial manuscript respectively (if different from the corresponding author) to participate. Authors with available telephone contact details were invited to complete the survey via a telephone interview. Prior to contacting via telephone, we emailed

an invitation attaching a study information sheet, a summary of the survey topics to be covered, and an opt-out form. Those authors without telephone contact details were contacted via email and sent the same information and a link to complete the same survey online via RED-Cap, a web survey hosting service [20]. Up to three reminder emails were sent to non-responders at intervals of approximately four weeks.

## Measures

**Suppression events.**   We asked respondents seven questions concerning their experiences when disseminating the trial results (see Box 1). The questions, based on those used by Yazahmeidi and Holman (2007) [14], had response options "not at all", "a little", or "substantially".

**Trial characteristics.**   Two researchers (SG and KM) independently extracted the following information from published reports of eligible trials: year of publication, the health risk behaviour(s) targeted (physical activity, nutrition, sexual health, substance use), and the country of first author, where the trial was assumed to have occurred. We aggregated author country into the categories: North America, Europe, Oceania, and Other.

A democracy classification was added for each publication based on the country of origin and the year of study publication using the Economist Intelligence Unit (EIU) Democracy Index reports [21–29]. The democracy index is a measure of a country's democracy and is based on five categories of 60 indicators which are scored to provide a total out of 10. Based on the score, countries are categorised into full democracy (score = 8.01 to 10), or not a full democracy (0 to 8). (N.B. there were no reports for the years 2007 and 2009 so this data is missing for studies published in those years).

While the focus of the study is government funding, we extracted data from all eligible reports and classified them in the following mutually exclusive categories: Dedicated Research agency (government), other Government Agency, Industry, and Philanthropic (see Box 2 for definitions). If funding information was unavailable, we coded the data as Unknown. Where

Box 1. Options respondents were provided regarding funder behaviour

• Reluctance to publish the findings in peer-reviewed journals as they were viewed as being unfavourable.

• Delays in reporting or publishing the findings until a more favourable time (e.g. following elections, after certain policies had been approved).

• Asked to alter your conclusions so that the impact of the intervention was framed in a way that aligned more with their interests.

• Asked to suppress certain findings as they were viewed as being unfavourable.

• Discouraged from presenting your results to certain groups or organisations that may have an interest in the intervention.

• Attempts to discredit members of the research team or other staff involved in the conduct of the study.

• Changes made to study methods or analytical procedures that would have likely resulted in an outcome that aligned more with their interests (e.g. significant finding).

Box 2. Definitions of funding categories

Dedicated research agency: A government funded agency solely responsible for medical and public health research.

Other government agency: A government agency, dedicated to pursuits other than research, including local councils, public health and safety departments, and ministerial departments.

Industry: Companies and activities involved in the production of goods for sale.

Philanthropic: A non-government, non-profit organisation, with assets provided by donors and managed by its own officials and with income expended for socially useful purposes.

Unknown: No funding source listed.

Multiple: Reports more than one of the previous funding type.

there was more than one source of funding, we coded the study as Multiple and excluded it from the regression analysis to avoid problems of attribution.

## Analysis

For each of the seven questions we calculated the proportion (aim 1) who answered "not at all" (coded as "never") versus "a little" or "substantially" (coded as "at least once"), and then a dichotomous variable indicating the proportion who had experienced any one of the seven suppression events. We conducted a sensitivity analysis to estimate the extremities of possible non-response bias, by assuming (a) that all non-respondents had experienced an act of suppression, and conversely, (b) that all non-respondents had not experienced an act of suppression.

We estimated associations between trial characteristics including the risk behaviour targeted, funder, geographic location, full democracy (yes vs no), and age of the publication (in years) and instances of suppression (aim 2) using logistic regression, recoding year of publication as the continuous variable 'age of publication in 2017' (the last year in the sampling frame). We estimated adjusted odds ratios with 95% confidence intervals, and aggregated trials where groups were small: 'sexual health/substance abuse (risky behaviour)' and 'nutrition/physical activity' based on evidence that these behaviours cluster [30, 31].

## Results

From 42 eligible reviews we identified 208 trials and received survey responses from 104 (50%) of their corresponding authors. Papers were published from 2007–2016 and reported trials concerning physical activity and/or nutrition (55%), substance use and/or sexual health (47%). Two thirds were conducted in North America (34%) or Europe/United Kingdom (33%), with the balance in Oceania (17%) and other countries (16%). Examining democracy data, the majority of studies were from full democracy countries (61%). The majority of studies receive funding from Other Government agencies (39%).

S1 Table shows that the characteristics of trials whose authors did not complete the survey were not markedly different from those who did, in terms of the study design, full democracy,

**Table 1. Researcher reports of funder efforts to suppress trial findings.**

| | Never | Once or more often* | | | | | |
|---|---|---|---|---|---|---|---|
| Funder Type | | *Industry* | *Other Government Agency* | *Philanthropic* | *Dedicated Research Agency* | *Multiple* | *Unknown* |
| Funder expressed reluctance for publication because they considered the results 'unfavourable' | 89 | 0 | 6 | 0 | 2 | 1 | 0 |
| Funder delayed reporting of findings until a more favourable time (e.g., following elections) | 93 | 0 | 2 | 0 | 0 | 2 | 1 |
| Funder asked researcher to alter conclusions to better align with funder interests | 91 | 0 | 3 | 0 | 1 | 2 | 0 |
| Funder asked researcher to not report findings they considered unfavourable | 95 | 0 | 2 | 0 | 1 | 0 | 0 |
| Funder discouraged researcher from presenting results to certain groups or organisations that may have an interest in the intervention | 95 | 0 | 1 | 0 | 1 | 1 | 0 |
| Funder attempted to discredit members of the research team or other staff involved in the conduct of the study | 94 | 0 | 1 | 0 | 2 | 1 | 0 |
| Funder demanded changes to study methods or analysis likely to produce findings that aligned with funder interests (e.g. emphasis on the "statistical significance" of a result) | 94 | 0 | 1 | 0 | 1 | 0 | 0 |

*Six respondents did not answer any of these questions, while the number of respondents answering each question ranged from 96 to 98.

or publication date. However, the proportion conducted in North America was higher among non-respondents (53%) than respondents (34%). Too, non-responder reported more funding from an Other Government Agency (49% non-completers vs 39% completers).

## Aim 1: Prevalence of suppression events

Eighteen percent (18/98, 6 unknown) of respondents reported at least one instance of suppression. Table 1 shows the number of respondents who reported each type of suppression event having occurred at least once, by funding source. Rates of suppression were highest for studies funded by Other Government agencies. The most commonly reported suppression event was that of the funder expressing reluctance for publication due to 'unfavourable' results: with six, two and one suppression events being reported from studies funded by other Government agencies, independent sources, and multiple funding sources, respectively. In comparison, researchers receiving industry or philanthropic funding did not report a single suppression event.

**Sensitivity analysis.** Under the extreme assumptions that none or all the non-respondents had experienced a suppression event, the prevalence estimates would be as low as 9% (18/208) or as high as 59% [(18+104)/208], respectively.

## Aim 2: Association between trial characteristics and suppression events

Table 2 summarises associations between trial characteristics and suppression events. Researchers receiving Other Government Grants, or who conducted studies in Europe, had higher odds of reporting a suppression event compared to those reporting on studies with independent dedicated research funding or conducting studies in North America, respectively. Researchers who had conducted sexual health/substance use trials more commonly reported a suppression event than those who had conducted nutrition/physical activity trials. Whether the publication came from a democratic country or not did not seem to change the odds of reporting suppression. As the age of the publication increased, so too did the odds of reporting

**Table 2. Associations between trial characteristics and suppression events.**

| | Reported a suppression event n (%) | Unadjusted Odds Ratio (95% CI) | Adjusted* Odds Ratio (95% CI) |
|---|---|---|---|
| Risk behaviour targeted | | | |
| Nutrition/physical activity | 7 (17%) | Ref | Ref |
| Sexual health/substance use | 8 (24%) | 1.49 (0.48, 4.65) | 2.25 (0.43, 11.68) |
| Funder | | | |
| Dedicated Research Agency | 5 (19%) | Ref | Ref |
| Philanthropic | 0 (0%) | - | - |
| Other Government Agency | 9 (28%) | 1.72 (0.50,5.95) | 2.22 (0.41, 12.10) |
| Industry | 0 (0%) | - | - |
| Unknown | 1 (10%) | 0.49 (0.05, 5.95) | - |
| Geographic location | | | |
| North America | 6 (20%) | Ref | Ref |
| Europe | 5 (21%) | 1.10 (0.29, 4.14) | 1.66 (0.28, 9.92) |
| Oceania | 2 (22%) | 1.19 (0.02, 7.25) | 0.59 (0.04, 8.36) |
| Other | 2 (18%) | 0.93 (0.16, 5.45) | 1.71 (0.04, 75.39) |
| Full democracy | | | |
| No | 2 (18%) | Ref | Ref |
| Yes | 9 (19%) | 1.04 (0.19–5.65) | 0.99 (0.03, 32.94) |
| Age of publication in 2017 (years), mean (standard deviation) | 6.73 (2.34) | 1.09 (0.86, 1.37) | 1.08 (0.79 to 1.49) |

*Adjusted for behaviour targeted, funder type, geographic location, and age of publication.

suppression. The confidence intervals for the odd ratios of all the comparisons were wide and included 1.

## Discussion

In a sample of authors of prevention intervention trial reports published over a decade, of whom 50% responded, we found that one in five reported at least one suppression event. A simple sensitivity analysis suggests that rates of suppression could be as low as 9%, and as high as 59%, depending on the proportion of non-respondents who were subjected to suppression events. Our overall estimate of 18% is similar to estimates from previous studies in Australia (21% to 34%) [15, 32] and Canada (24%) [16]. Notably, we asked specifically about what occurred in relation to a single trial, while other studies evaluated suppression events across many projects over the course of five [16] or 5.5 years [14]. As such, rates of suppression may be much higher.

The sampling frame provides for greater international representation and broader coverage of health research than previous studies [14, 32]. However, relying on published studies to direct us to authors means that we would not have identified authors of studies whose publication was supressed entirely. It is also likely that some authors would not disclose suppression events, even in the confidential context of a research study, fearing repercussions from the funder, or being negatively evaluated by the researchers. The implication of this being that the actual rate of suppression is much higher. Finally, the small numbers in our study constrain the precision of our estimates of prevalence and association. Accordingly, we suggest that 18% is an under-estimate of the true prevalence of studies subject to some form of suppression by funders.

It is hard to determine why older publications or those published in different geographical regions were suggestive of having to have greater odds of suppression (though the confidence intervals for all the comparisons were wide and included 1). A possible explanation may be that older studies which have existed longer may have had more 'chance' to experience suppression on their findings. That studies published outside North America were found to have greater odds of suppression is hard to elucidate. It may be that instances of suppression in North America are only reported by individual researchers when it is 'more severe' than those reported in other countries, that they were more afraid to report suppression, or that there are tighter regulations on what types of suppression can be enacted on grant holders. More research would be needed to explain the differences noted.

Our results, along with those of previous investigations, suggest that government funders interfere with public-good research. In addition to curtailing independent scientific enquiry, such practices deny the public access to the findings of research paid for through taxation, which in some cases, could have informed policy decisions. On this point, in his 'Missing Evidence' report, former High Court Judge Sir Stephen Sedley observed that the UK Parliament made a critical decision concerning the merits of minimum unit pricing of alcohol without the benefit of key findings whose publication had been deliberately delayed by the Department of Health [4]. In addition to the loss borne by taxpayers due to ill-informed policy, is the damage done to democracy when such perversions come to light.

This research has its limitations. For example, we did not attempt to 'deep dive' on the different types of suppression. We found that reluctance to publish research findings due to unfavourable results was the most reported type of suppression experiences. However, we did not seek to determine what reluctance meant in this context–whether it meant funders prevented publication or just tried to influence it. Too we did not give participants the opportunity to describe other forms of suppression they may have felt. Further research should investigate this in order to determine all type of suppression experienced in order to develop the most practical ways to overcome them.

Attention is urgently needed to protect the integrity of public health research from the influence of vested interests, whether private or official in origin. Preventive actions are required of all actors involved in the generation of research findings:

1. Government agencies must ensure appropriate terms in funding agreements formed with research providers that protect academic freedom for e.g. the removal of clauses which require the approval of results prior to publication [32];

2. Research institutions must not accept funding on terms that permit funders to interfere in public-good research;

3. Government agencies should establish a registry of studies funded by government agencies including the terms of the funding to encourage openness;

4. Research Ethics Committees must consider the source and terms of research funding to determine if there are any ethical implications of the funding source;

5. Scientific journals must require authors to declare the terms of research funding, and potential conflicts of interest;

6. Researchers must be held to Code of Conduct provisions concerning acceptable terms of funding;

7. Audits of contracting and research practices of tertiary academic institutions should be undertaken by an independent body with appropriate powers (e.g. an independent government department); and

8. Universities should consider establishing a mechanism for reporting instances research suppression, and the management of funders/individuals/etc. who are known to attempt to suppress research findings.

Suppression still exists among public-good researchers, with this study suggesting rates of suppression for a single trial may be as high as one in five. Prevention is key and these suggestions, similar to those previously described [32], need to be adopted in order to thwart the occurrence of suppression of public-good health research. As it is unlikely that instances of suppression will ever truly be stopped, further research needs to be conducted to determine ways of handling suppression when they do happen at the researcher level (e.g. what can a researcher do if someone tries to delay their publication), as well as reporting procedures in place if instances of suppression do occur.

## Supporting information

**S1 Table. Characteristics of responders and non-responders.**
(DOCX)

**S1 File.**
(XLSX)

**S2 File.**
(XLSX)

**S3 File.**
(XLSX)

**S4 File.**
(XLSX)

## Author Contributions

**Conceptualization:** Sam McCrabb, Luke Wolfenden, Kypros Kypri.

**Formal analysis:** Sam McCrabb, Luke Wolfenden, Kypros Kypri.

**Funding acquisition:** Luke Wolfenden.

**Investigation:** Sam McCrabb, Luke Wolfenden, Kypros Kypri.

**Methodology:** Kaitlin Mooney, Sharleen Gonzalez, Elizabeth Ditton, Serene Yoong.

**Writing – original draft:** Sam McCrabb, Kypros Kypri.

**Writing – review & editing:** Kaitlin Mooney, Luke Wolfenden, Sharleen Gonzalez, Elizabeth Ditton, Serene Yoong.

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
