## [Decision Letter · Decision Letter 0]

22 Jun 2021

PONE-D-21-13007

"He who pays the piper calls the tune": Researcher experiences of funder suppression of health behaviour intervention trial findings

PLOS ONE

Dear Dr. McCrabb,

Thank you for submitting your manuscript to PLOS ONE. After careful consideration, we feel that it has merit but does not fully meet PLOS ONE’s publication criteria as it currently stands. Therefore, we invite you to submit a revised version of the manuscript that addresses the points raised during the review process.

Your manuscript was very well received by the reviewers and they largely note minor revisions for clarity. Please also address the outstanding queries regarding recruitment and some of the nuances in terms of interpretation (or limits of interpretation) regarding what suppression actually entails. One of the reviewers suggests a novel analysis by region/country, which I would ask you to consider or respond to. I look forward to receiving your revision.

We look forward to receiving your revised manuscript.

Kind regards,

Quinn Grundy, PhD, RN

Academic Editor

PLOS ONE

Journal Requirements:

Reviewers' comments:

Reviewer's Responses to Questions

**Comments to the Author**

1. Is the manuscript technically sound, and do the data support the conclusions?

Reviewer #1: Yes

Reviewer #2: Yes

Reviewer #3: Yes

2. Has the statistical analysis been performed appropriately and rigorously? 

Reviewer #1: I Don't Know

Reviewer #2: Yes

Reviewer #3: Yes

3. Have the authors made all data underlying the findings in their manuscript fully available?

Reviewer #1: No

Reviewer #2: No

Reviewer #3: Yes

4. Is the manuscript presented in an intelligible fashion and written in standard English?

Reviewer #1: Yes

Reviewer #2: Yes

Reviewer #3: Yes

5. Review Comments to the Author

Reviewer #1: This is an interesting and important paper but I have a number of suggestions for improvement. It reports the extent to which funders of studies attempt to suppress publication of unfavourable results, which is a highly important topic.

Major comments:

Recruitment part slightly unclear: “We invited the first two authors, the last author, and the corresponding author, using contact information in the public domain, to complete the survey. Authors with available contact details were invited to complete a telephone interview.” If they were contacted they all had contact details available – what does this last sentence mean? How was it actually decided whether to do interview or survey? Did you only do interview where phone number was available? Why not email to arrange phone interview and ask for number then? Were questions the same?

It would be helpful to know what “reluctance” actually signifies. Funders shouldn’t be able to affect whether findings are published – did responses suggest that funders could actually prevent publication or just tried to influence? A related point is that it would be very interesting to know how researchers dealt with suppression attempts – particularly attacks on the research team.

Generally the discussion (which is rather short) and/or results would benefit from more detail on the details of suppression attempts and how they were handled, if these data are available.

Minor comments:

Abstract should mention other aspects apart from supression specifically to give readers an idea of other behaviours.

Some minor lack of clarity in places, eg. “improve smoking/substance abuse ”

Why these specific health intervention areas and not others? A rationale should be given.

Were respondents able to suggest other types of suppression activity? There may be other types not mentioned in your survey.

Reviewer #2: This is an excellent and timely paper that is well written and comes to an important and well justified set of recommendations. I have only a few minor comments that will clarify the presentation and improve the general context of the research and interpretation.

I think it would be more usual to put the . after the references, such as [1, 2].

77-83. You could further improve the context by referring to the recent study on science suppression in ecology by Driscoll et al. 2020 in Conservation Letters, relevant because it addresses suppression in public good research in Australia.

98. researchers'

203. insert "the odds ratios of" at the start of this line.

211-12. Is it worth reporting these extreme values, as neither is likely to be true, or even close to true?

224-5. That is a reasonable interpretation.

226. Hold on, you said at 201-2 that older studies had higher rates of suppression. Some extra words in the results to help interpret direction of effect in relation to the odds ratio would help readers and the authors to avoid this mix up.

228. It's a bit confusing to combine older and north American in this sentence, giving the impression it was only older north American studies, but these tests were independent, so it's old studies everywhere, and any aged study in north America.

235-6; or north American researchers are more afraid to report suppression. Or you could look at the European countries that your studies came from to see if they include less democratic countries. In fact, in addition to the regional analysis, it would be interesting to examine the likelihood of suppression in relation to the global democracy index, which extends back to 2006, so could be aligned with the year each paper was published, or the year before.

251; spell out what that would mean, what kind of clauses must be excluded from contracts or included for example.

Also, in this list of actions, be clear about who should it for each point. It might also refer to other literature where similar lists of actions have been referred to (this might be best done in the list or as an introduction to the list).

It would be nice to wrap up with a general, bigger picture paragraph about interference in public good research.

I couldn't see if the data were available, which Plos seems to require.

Reviewer #3: This was a very clear, concise and well-presented paper. It highlights a key aspect of research suppression, namely pressure from funders to produce and publish findings that align with the priorities of the day.

It was necessarily narrow in scope. However, it would be interesting to know whether respondents experienced suppression events from parties other than the funding body. In my experience, public health fields are particularly prone to silencing from within. It also would have been interesting to include discussion of the broader, more insidious suppression that chronically underfunded public research produces. As noted in the authors' limitations section, it is impossible to ascertain the number of researchers whose work is suppressed before it has even begun. The unspoken or whispered warnings to keep one's head down; stick to something safer; only 'pick winners'. We know in Australia for example, that it is not worth writing a grant funding proposal that suggests the government's alcohol guidelines are not conducive to responsible drinking, or that school-based fitness programs may actually be counter-productive.

The only suggestions I would make are:

-Page 5, line 98 contains a missing apostrophe. It should read "researchers' experiences"

-Page 6, lines 129-130, title of Box 1 is convoluted. Consider renaming to “Options respondents were provided regarding funder behaviour”

6. PLOS authors have the option to publish the peer review history of their article (what does this mean?). If published, this will include your full peer review and any attached files.

Reviewer #1: No

Reviewer #2: No

Reviewer #3: **Yes: **Dr Jacqueline Hoepner

---

## [Author Response · Author response to Decision Letter 0]

19 Jul 2021

16 July 2021

Dear Dr Quinn Grundy

PLOS ONE

Thank you for your email dated 23rd June 2021 and the opportunity to revise and resubmit our manuscript entitled ‘"He who pays the piper calls the tune": Researcher experiences of funder suppression of health behaviour intervention trial findings’ (PONE-D-21-13007). Please find below our response to reviewers. 

Reviewer #1: This is an interesting and important paper but I have a number of suggestions for improvement. It reports the extent to which funders of studies attempt to suppress publication of unfavourable results, which is a highly important topic.

Major comments:

1. Recruitment part slightly unclear: “We invited the first two authors, the last author, and the corresponding author, using contact information in the public domain, to complete the survey. Authors with available contact details were invited to complete a telephone interview.” If they were contacted they all had contact details available – what does this last sentence mean? How was it actually decided whether to do interview or survey? Did you only do interview where phone number was available? Why not email to arrange phone interview and ask for number then? Were questions the same?

We have rephrased the text around recruitment. Individuals were surveyed via telephone if they had a contact number available, if not, they were contacted via email and asked to complete the same survey online. We have attempted to clarify in the text. 

“Authors from identified articles were invited to participate if they were one of the first two authors, the last author, or the corresponding author. Contact information was sourced from the public domain. We contacted corresponding authors first to complete the survey on behalf of all authors. Corresponding authors could nominate co-authors to complete the survey on their behalf. If after four week we had no response to the corresponding author, we invited the first, second and/or last author of the trial manuscript (if different from the corresponding author) to participate. Authors with available telephone contact details were invited to complete the survey via a telephone interview. Prior to contacting via telephone, we emailed an invitation attaching a study information sheet, a summary of the survey topics to be covered, and an opt-out form. Those authors without telephone contact details were contacted via email and sent the same information and a link to complete the same survey online via REDCap, a web survey hosting service.[19] Up to three reminder emails were sent to non-responders at intervals of approximately four weeks.”

2. It would be helpful to know what “reluctance” actually signifies. Funders shouldn’t be able to affect whether findings are published – did responses suggest that funders could actually prevent publication or just tried to influence? A related point is that it would be very interesting to know how researchers dealt with suppression attempts – particularly attacks on the research team.

3. Generally the discussion (which is rather short) and/or results would benefit from more detail on the details of suppression attempts and how they were handled, if these data are available.

We respond to comments 2 and 3 together. Unfortunately this information is not available. We have added to the discussion that this is a limitation of our research, and that further research would benefit from investigating. 

“This research has its limitations. For example, we did not attempt to ‘deep dive’ on the different types of suppression. We found that reluctance to publish research findings due to unfavourable results was the most reported type of suppression experiences. However, we did not seek to determine what reluctance meant in this context – whether it meant funders prevented publication or just tried to influence it. Further research should investigate this in order to determine the most practical ways to overcome these types of suppression.”

Minor comments:

4. Abstract should mention other aspects apart from suppression specifically to give readers an idea of other behaviours.

We have added additional information to the abstract to indicate other parts of the study.

“We also collected information on study funder, geographical location and age of publication and how these were associated to reported suppression.”

5. Some minor lack of clarity in places, e.g. “improve smoking/substance abuse”

We have edited the text thoroughly to improve expression and clarity.

6. Why these specific health intervention areas and not others? A rationale should be given.

As this is part of a larger study examining public health primary prevention researchers’ experiences in developing, conducting and evaluating public health interventions, the health intervention areas where selected as areas of interest.

We have added to the text “These risk behaviour areas were chosen as the larger cross-sectional study this study is a part of was interested in these health behaviour domains.”

7. Were respondents able to suggest other types of suppression activity? There may be other types not mentioned in your survey.

We did not give participants the opportunity to describe other forms of suppression. We have added this to the limitation section in the discussion. 

“Too we did not give participants the opportunity to describe other forms of suppression they may have felt.”

Reviewer #2: This is an excellent and timely paper that is well written and comes to an important and well justified set of recommendations. I have only a few minor comments that will clarify the presentation and improve the general context of the research and interpretation.

8. I think it would be more usual to put the . after the references, such as [1, 2].

Thank you for your suggestion. We follow AMA in-text referencing guidelines which places references outside of periods and commas so have not updated the text for this suggestion.

9. 77-83. You could further improve the context by referring to the recent study on science suppression in ecology by Driscoll et al. 2020 in Conservation Letters, relevant because it addresses suppression in public good research in Australia.

Thank you for your suggestion. We have added this reference to the text

“A survey of Australian ecologists and conservation scientists by Driscoll et al. indicated that Government and industry respondents reported higher rates of suppression than university respondents (34%, 30% and 5% respectively) and this suppression was mainly in the form of internal communication and media.”

10. 98. researchers' & 203. insert "the odds ratios of" at the start of this line.

We have updated the text for these suggestions.

11. 211-12. Is it worth reporting these extreme values, as neither is likely to be true, or even close to true? 224-5. That is a reasonable interpretation.

Given the large number of non-respondents for the survey (50%) we believe it add context to the range of potential suppression which may have been reported by respondents. This additional analysis add supports to our discussion where we highlight that we believe the reported prevalence of 18% is an underrepresentation which you highlight is a reasonable interpretation.

12. 226. Hold on, you said at 201-2 that older studies had higher rates of suppression. Some extra words in the results to help interpret direction of effect in relation to the odds ratio would help readers and the authors to avoid this mix up. 228. It's a bit confusing to combine older and north American in this sentence, giving the impression it was only older north American studies, but these tests were independent, so it's old studies everywhere, and any aged study in north America.

Thank you for highlighting this oversight we have updated the text.

“It is hard to determine why older publications or those published in different geographical regions were suggestive of having to have greater odds of suppression compared to younger publications published in North America (though the confidence intervals for all the comparisons were wide and included 1). A possible explanation may be that older studies which have existed longer may have had more chance to experience suppression on their findings.”

13. 235-6; or north American researchers are more afraid to report suppression. Or you could look at the European countries that your studies came from to see if they include less democratic countries. In fact, in addition to the regional analysis, it would be interesting to examine the likelihood of suppression in relation to the global democracy index, which extends back to 2006, so could be aligned with the year each paper was published, or the year before.

We have added the variable full democracy (yes, no) to our analysis based on the Global democracy index as suggested (please see Table 2). This analysis did not highlight any differences between democratic countries and others. 

14. 251; spell out what that would mean, what kind of clauses must be excluded from contracts or included for example.

We have added an example of the types of clauses we believe should be omitted to the text. 

“Government agencies must ensure appropriate terms in funding agreements formed with research providers that protect academic freedom for e.g. the removal of clauses which require the approval of results prior to publication”

15. Also, in this list of actions, be clear about who should it for each point. It might also refer to other literature where similar lists of actions have been referred to (this might be best done in the list or as an introduction to the list). It would be nice to wrap up with a general, bigger picture paragraph about interference in public good research.

We have added additional information to the checklist and a summary paragraph to the end of the manuscript.

“Suppression still exists among public-good researchers, with this study suggesting rates of suppression for a single trial may be as high as one in five. Prevention is key and these suggestions, similar to those previously described,{Kypri, 2015 #68} need to be adopted in order to thwart the occurrence of suppression of public-good health research. As it is unlikely that instances of suppression will ever truly be stopped, further research needs to be conducted to determine ways of handling suppression when they do happen at the researcher level (e.g. what can a researcher do if someone tries to delay their publication), as well as reporting procedures in place if instances of suppression do occur.”

16. I couldn't see if the data were available, which Plos seems to require.

We have added this data to the submission.

Reviewer #3: This was a very clear, concise and well-presented paper. It highlights a key aspect of research suppression, namely pressure from funders to produce and publish findings that align with the priorities of the day.

It was necessarily narrow in scope. However, it would be interesting to know whether respondents experienced suppression events from parties other than the funding body. In my experience, public health fields are particularly prone to silencing from within. It also would have been interesting to include discussion of the broader, more insidious suppression that chronically underfunded public research produces. As noted in the authors' limitations section, it is impossible to ascertain the number of researchers whose work is suppressed before it has even begun. The unspoken or whispered warnings to keep one's head down; stick to something safer; only 'pick winners'. We know in Australia for example, that it is not worth writing a grant funding proposal that suggests the government's alcohol guidelines are not conducive to responsible drinking, or that school-based fitness programs may actually be counter-productive.

The only suggestions I would make are:

-Page 5, line 98 contains a missing apostrophe. It should read "researchers' experiences"

Thank you, we have updated this in the text.

-Page 6, lines 129-130, title of Box 1 is convoluted. Consider renaming to “Options respondents were provided regarding funder behaviour”

Thank you, we have updated this in the text.

We hope you find these modifications and explanations satisfactory. We hope to be considered for publication in PLOS ONE. 

Kind regards,

Dr Sam McCrabb

School of Medicine and Public Health

Faculty of Health and Medicine

---

## [Editor Report · Decision Letter 1]

22 Jul 2021

"He who pays the piper calls the tune": Researcher experiences of funder suppression of health behaviour intervention trial findings

PONE-D-21-13007R1

Dear Dr. McCrabb,

We’re pleased to inform you that your manuscript has been judged scientifically suitable for publication and will be formally accepted for publication once it meets all outstanding technical requirements.

Kind regards,

Quinn Grundy, PhD, RN

Academic Editor

PLOS ONE
---

## [Editor Report · Acceptance letter]

26 Jul 2021

PONE-D-21-13007R1 

“He who pays the piper calls the tune”: Researcher experiences of funder suppression of health behaviour intervention trial findings 

Dear Dr. McCrabb:

I'm pleased to inform you that your manuscript has been deemed suitable for publication in PLOS ONE. Congratulations! Your manuscript is now with our production department. 

Kind regards, 

on behalf of

Dr. Quinn Grundy 

Academic Editor

PLOS ONE